# Randomized Clinical Trial of How Long-Term Glutathione Supplementation Offers Protection from Oxidative Damage and Improves HbA1c in Elderly Type 2 Diabetic Patients

**DOI:** 10.3390/antiox11051026

**Published:** 2022-05-23

**Authors:** Saurabh Kalamkar, Jhankar Acharya, Arjun Kolappurath Madathil, Vijay Gajjar, Uma Divate, Sucheta Karandikar-Iyer, Pranay Goel, Saroj Ghaskadbi

**Affiliations:** 1Department of Zoology, Savitribai Phule Pune University, Pune 411007, India; sdkalamkar@unipune.ac.in (S.K.); jhankaracharya@unipune.ac.in (J.A.); 2Biology Division, Indian Institute of Science Education and Research, Pune 411008, India; k.marjun@students.iiserpune.ac.in; 3Department of Medicine, Jehangir Hospital, Pune 411001, India; vijaygajjar2008@gmail.com; 4Jehangir Clinical Development Centre, Pune 411001, India; umadivate@jcdc.co.in; 5Iyer Clinic, Pune 411030, India; academics@dmhospital.org

**Keywords:** GSH supplementation, type 2 diabetes, HbA1c, oxidative stress, 8-OHdG, elderly diabetic population

## Abstract

Complications in type 2 diabetes (T2D) arise from hyperglycemia-induced oxidative stress. Here, we examined the effectiveness of supplementation with the endogenous antioxidant glutathione (GSH) during anti-diabetic treatment. A total of 104 non-diabetic and 250 diabetic individuals on anti-diabetic therapy, of either sex and aged between 30 and 78 years, were recruited. A total of 125 diabetic patients were additionally given 500 mg oral GSH supplementation daily for a period of six months. Fasting and PP glucose, insulin, HbA1c, GSH, oxidized glutathione (GSSG), and 8-hydroxy-2-deoxy guanosine (8-OHdG) were measured upon recruitment and after three and six months of supplementation. Statistical significance and effect size were assessed longitudinally across all arms. Blood GSH increased (Cohen’s d = 1.01) and 8-OHdG decreased (Cohen’s d = −1.07) significantly within three months (*p* < 0.001) in diabetic individuals. A post hoc sub-group analysis showed that HbA1c (Cohen’s d = −0.41; *p* < 0.05) and fasting insulin levels (Cohen’s d = 0.56; *p* < 0.05) changed significantly in diabetic individuals above 55 years. GSH supplementation caused a significant increase in blood GSH and helped maintain the baseline HbA1c overall. These results suggest GSH supplementation is of considerable benefit to patients above 55 years, not only supporting decreased glycated hemoglobin (HbA1c) and 8-OHdG but also increasing fasting insulin. The clinical implication of our study is that the oral administration of GSH potentially complements anti-diabetic therapy in achieving better glycemic targets, especially in the elderly population.

## 1. Introduction

Hyperglycemia causes micro- and macrovascular complications in type 2 diabetes (T2D) through oxidative stress. This is mediated by the overproduction of reactive oxygen species (ROS) through four pathways, namely advanced glycation end products, polyol, hexosamine, and protein kinase C [1]. Animal studies have shown that scavenging hyperglycemia-mediated ROS with antioxidants such as N-acetyl-cysteine (NAC), lipoic acid, and glutathione (GSH), or precursors of GSH, such as glycine and cysteine [2,3,4,5], not only partially improved blood glucose levels, the functionality of β-cells, and insulin sensitivity, but also reduced diabetic complications. However, there are few human studies that confirm the role of antioxidants as a potential supplementary treatment in diabetes.

Clinical trials of GSH supplementation, in particular, have received a great deal of attention. GSH is an endogenous antioxidant necessary to detoxify free radicals and maintain the redox homeostasis of the cell. Low levels of GSH are associated with many pathological conditions, such as cancer, arthritis, cardiovascular and neurodegenerative diseases, and diabetes [6]. Several reports, including work from our lab, have confirmed that GSH levels are significantly lower in diabetic patients [7,8,9], and controlling hyperglycemia over a period of two months increases blood GSH levels and reduces oxidative damage significantly [8]. The compensation of GSH insufficiency through supplementation may help in further arresting the development of complications in T2D by improving the redox state.

GSH has been orally administered in forms such as sublingual [10], orobuccal [11,12], and liposomal [13] for rapid absorption. We note that these forms of GSH are not only not easily available commercially but also sublingual and orobuccal formulations include GSH as one of the (primary) ingredients, which makes it difficult to attribute the effects to GSH alone. Richie et al. (2015) [14] demonstrated that oral GSH supplementation in 20 healthy individuals led to a significant increase in blood GSH. In a somewhat larger study conducted on 40 healthy American adults, however, Allen and Bradly (2011) [15] reported that oral GSH supplementation did not change GSH levels and biomarkers of oxidative stress. Precursor amino acids of GSH administered orally have also demonstrated enhanced body stores of GSH [16] in humans. Sekhar et al. [17] showed that dietary supplementation with cysteine and glycine, precursors of GSH, increased the rate of GSH synthesis and reduced lipid peroxidation in 12 American diabetic individuals without any change in glycated hemoglobin (HbA1c). They claimed that the deficient synthesis of GSH was restored by the oral supplementation with cysteine and glycine in eight older patients, but not in young individuals [17]. GSH has an added advantage over its precursor amino acids, for instance, cysteine, which has an unpleasant taste, in ensuring better patient compliance. Paolisso et al. [18] reported that GSH infusion led to increased GSH and total body glucose disposal in 10 Italian diabetic subjects; this effect was more pronounced in elderly individuals with impaired glucose tolerance [19]. Infusion is clearly difficult to implement in clinical practice. Most of these clinical studies have been carried out with small sample sizes and are often inconclusive. Discrepancies in the outcomes of these studies could be due to differences in the dose and duration of GSH, and the site of measurement of GSH being plasma instead of an erythrocyte fraction. Moreover, while most of these studies have focused on restoring body stores of GSH in both healthy and diabetic individuals, few have reported their effects on alleviating oxidative stress, or for that matter, glycemic stress itself. A detailed summary of all these trials is provided in Appendix A.

Since we intended to measure HbA1c as a marker in our study (RBC lifespan is typically taken as 120 days), the overall study duration was chosen to be 6 months, which allowed two measurements of change in HbA1c, 3 months apart. This allowed us to establish long-term effects and study the stability of the observations to prolonged supplementation.

We conducted a pragmatic clinical trial prospectively in 200 Indian diabetic patients to assess whether supplementation with oral GSH improves body stores of GSH. We further asked if GSH supplementation for a relatively prolonged duration (six months) co-administered with ongoing anti-diabetic treatment supports glycemic control by minimizing oxidative damage. We serially measured concentrations of GSH, 8-hydroxy-2-deoxy guanosine (8-OHdG; an oxidative damage marker), and glycemic parameters in diabetic patients receiving GSH supplementation in addition to anti-diabetic treatment, and compared them with serial measurements in those receiving anti-diabetic treatment alone. Our study results show that oral GSH supplementation not only improved body stores of GSH and significantly reduced oxidative damage but also helped maintain lower HbA1c in elderly diabetic patients. We noted that this effect of GSH supplementation was more pronounced in elderly individuals.

## 2. Subjects, Materials, and Methods

### 2.1. Ethical Approval

This study was approved by the Institutional Ethical Committee of Jehangir Hospital Development Center, Pune (JCDC ECN- ECR/352/Inst/NIH/2013); Institutional Biosafety Committee of SPPU (Bot/27A/15), Pune; and the Institutional Ethical Committee of IISER, Pune (IECHR/Admin/2019/001). Signed informed consent was obtained from all the subjects at the time of enrollment in the study after explaining the purpose and nature of the study. All participants in this study were de-identified using a numbered code. This study is registered with the Clinical Trials Registry—India (CTRI/2018/01/011257). This study was conducted in compliance with CONSORT guidelines and guidelines of the Helsinki declaration.

### 2.2. Study Design

We conducted a pragmatic clinical trial designed as a case-control cohort study to assess the effect of oral GSH supplementation on blood GSH levels and glucose homeostasis in diabetic patients.

### 2.3. Inclusion/Exclusion Criteria for Study Participants

We recruited healthy non-diabetic controls (*n* = 104) with HbA1c < 6.5%, and known T2D subjects (*n* = 250) with HbA1c ≥ 6.5% [20] visiting Jehangir Hospital and Iyer clinic, Pune. Pregnant women, heavy smokers, individuals with excessive alcohol intake, individuals with any clinical infection or with a history of a recent cardiovascular event, and those receiving antioxidants or herbal formulations were excluded from the study. Body weight, height, anti-diabetic treatment, and family history of diabetes were noted for each subject.

### 2.4. Recruitment and Randomization for GSH Intervention

We recruited known diabetic subjects (*n* = 250) who were already on anti-diabetic regimen and study physician randomly categorized them into two groups based on coin-toss method: 125 diabetic patients were advised to continue with their anti-diabetic regimen (Group D), and the other 125 diabetic patients were given oral 500 mg glutathione (Jarrow Formulas, Los Angeles, CA, USA) supplementation once daily in addition to their anti-diabetic treatment (Appendix A) for a period of six months (Group DG) (Figure 1). At the time of randomization, concentrations of covariates, fasting and postprandial (PP) glucose and insulin, HbA1c, GSH and oxidized glutathione (GSSG), and 8-OHdG were not available, and therefore did not influence the assignment of diabetic patients in D or DG groups. Compliance to medical treatment by patients of D and DG group and consumption of GSH by patients of the DG was emphasized by maintaining continuous communication between the physician and patients. Out of 125 diabetic patients in D and DG group, 23 were lost to follow-up in the D group and 21 in DG group for not complying with the treatment regimen. We also recruited healthy non-diabetic control subjects who were followed for six months, during which they were advised to continue with their regular diet and exercise regimen. Blood samples were collected at the time of enrollment 0 (α visit), 3 (β visit), and 6 (γ visit) months after the date of enrollment.

#### 2.4.1. Sample Size Calculation

Sample Size (*n* = 100) is calculated based on a two-sided *t*-test, at 0.1 type 1 error and 80% power, to detect a mean difference of 35 in GSH with a standard deviation of 100.

#### 2.4.2. Sample Collection

At each visit, a total of 10 mL fasting and postprandial (PP) blood samples were collected from all the subjects at Golwilkar Metropolis, Pune. Blood samples were centrifuged at 4000 rpm for 10 min to separate erythrocyte fraction from whole blood. Plasma was stored at −80 °C (Thermo Fisher Scientific, Asheville, NC, USA) for further analysis.

### 2.5. Estimation of Blood Biochemical Parameters

Measurement of fasting plasma glucose (FPG), postprandial blood glucose (PPG), fasting plasma insulin (FPI), postprandial insulin (PPI), and HbA1c was performed on an automated analyzer at Golwilkar Metropolis, Pune, following CLSI (Clinical and Laboratory Standards Institute, Malvern, PA, USA) guidelines. Erythrocyte hemolysate was prepared by washing it twice with cold saline and hemolyzing by adding ice-cold ultrapure water [8]. This was stored at −80 °C for further analysis.

### 2.6. Estimation of GSH and GSSG

Reduced and total glutathione content in erythrocyte hemolysate was estimated using glutathione assay kit (Cayman Chemical, Ann Arbor, MI, USA). This kit follows DTNB (5,5′-dithio-bis-2 nitrobenzoic acid, Ellman’s reagent) method for estimation of GSH [21], where DTNB reacts with reduced GSH, yielding yellow-colored 2-nitro-5-thiobenzoate, which is read at 405 nm on ELISA reader. Briefly, 50 µL of erythrocyte lysate was deproteinized using an equal volume of metaphosphoric acid at 4 °C. After vigorous vortexing, the resulting mixture was centrifuged at 2000× *g* for 2 min at 4 °C. The supernatant was separated and aliquoted in two parts and used for estimation of total GSH (TGSH) and GSSG. The pH of the samples was adjusted to 8 by addition of triethanolamine (5 µL/100 µL of the sample). One of the aliquots was diluted 50 times with 1× MES buffer (0.4 M (N-morpholino) ethanesulphonic acid, 0.1 M phosphate buffer, and 2 mM EDTA ph 6) and used for estimation of TGSH. In the second aliquot, 2 µL of vinyl pyridine was added and diluted 25 times with 1× MES buffer and 50 µL of this sample was used for estimation of GSSG. Both the aliquots were then incubated for 1 h at room temperature. The reaction was started by adding 150 µL assay cocktail (11.25 mL MES (N-morpholino) ethanesulphonic acid, 0.1 M phosphate buffer, and 2 mM EDTA ph 6) buffer, 0.45 mL cofactor mixture containing NADP^+^ and glucose-6-phosphate, 2.1 mL enzyme mixture containing glutathione reductase and glucose-6-phosphate dehydrogenase, 2.3 mL water, and 0.45 mL DTNB. Increase in TNB formation was determined by measuring absorbance at 405 nm at 5 min interval for 30 min. GSSG was used as a standard for estimating the concentration of TGSH and GSSG in samples. Absorbance values of samples and standard (0, 0.5, 1.0, 2.0, 4.0, 8.0, 12.0, and 16.0 µM) were plotted as a function of time, and slope for each sample was calculated. This was called i-slope. The i-slope of each concentration of standard was plotted against the concentration of GSH and the slope of this curve was called f-slope. Values for total GSH and GSSG were calculated by using the formula given below. GSH concentration was determined by subtracting GSSG from total GSH.
$$\mathrm{GSH}\;(µM)=\frac{\left(i-\mathrm{slope}\;\mathrm{for}\;\mathrm{sample}\right)-\left(y\;\mathrm{intercept}\right)\;}{f-\mathrm{slope}}\;\times \;\mathrm{sample}\;\mathrm{dilution}$$

### 2.7. Estimation of 8-OHdG

DNA was isolated from whole blood by standard phenol–chloroform isoamyl alcohol extraction method and quantitated on nanodrop. Amount of 8-OHdG in DNA was determined by competitive enzyme-linked immunosorbent assay using standard protocol of Modak et al. (2009) [22]. Briefly, 96-well plate was coated with 100 µL 0.003% protamine sulphate (Sigma, St. Louis, MO, USA) and then incubated at 37 °C for 5–6 h. Protamine sulphate solution was removed and 100 ng of 8-OHdG was added to each well and incubated at 4 °C overnight. The plate was washed with phosphate-buffered saline (PBS) and incubated with monoclonal antibody against 8-OHdG (1 mg/mL) (1:5000) already mixed with either standard 8-OHdG or experimental DNA samples and incubated for 3–4 h at 37 °C. Experimental samples consisted of 100 ng genomic DNA of the individuals from three study groups (C, D, and DG) at α, β, and γ visits. After washing the plate with PBS containing Tween-20 (PBST) 5 times, it was incubated with 100 µL (1:2500) of goat anti-mouse antibody conjugated with a biotin FAb fragment per well at 37 °C for 30 min. The plate was then washed 5 times with PBST and incubated with 100 µL (1:5000) of avidin conjugated with horseradish peroxidase enzyme at 37 °C for 30 min. Finally, after 3 washings of PBST and 3 washings of phosphate citrate buffer, pH 5, 100 µL of ABTS substrate solution containing 0.06% H_2_O_2_ was added to each well incubated for 10 min and the absorbance was measured at 405 nm using Multiskan plate reader (Thermo scientific, Shanghai, China) and expressed as ng 8-OHdG/µL DNA.

Statistical Methods

Biochemical parameters of subjects in Control, D, and DG groups at the first visit were represented using the descriptive statistics (Median, 25th percentile, and 75th percentile). All intra- and inter-group comparisons of biochemical parameters at different visits were performed using permutation tests, using the “Coin” package in R [23]. Statistical significance was set at *p*-value < 0.05. The results of permutation tests were confirmed with two-sample, two-sided *t*-tests. The results obtained from permutation tests were presented here. Effect size analysis was used to quantify the difference between 6-month biochemical changes in D and DG groups. All calculations and parametric *t*-tests were carried out using Matlab version 2019.

Effect Size Calculations

Biochemical measurements of variables are available at α, β, and γ visits. Changes in the biochemical variables, HbA1c, fasting glucose (FPG), fasting insulin (FPI), PP glucose (PPG), PP insulin (PPI), GSH, GSSG, and 8-OHdG in during the study period from α to γ visit (6 months) in a group were estimated by taking their paired differences between those visits. Let the 6-month changes in D for a variable *x* be denoted by $\;D_{x}$, and similarly for DG group by $\;DG_{x}$. The effect size of 6-month changes in the concentration of a particular biochemical variable *x* (*x* can be HbA1c, FPG, FPI, PPG, PPI, GSH, GSSG, and 8-OHdG) between D and DG groups is estimated using Cohen’s d [24] as
$$d=\frac{\;mean\;of\;DG_{x}-mean\;of\;D_{x}}{s}$$
where, *s* is the pooled standard deviation of changes in the *x* variable in D and DG groups. Cohen (1969) described an effect size of 0.2, 0.5, and 0.8 as “Small”, “Medium”, and “Large” effects respectively, and Sawilowsky [25] classified an effect of sizes 1.2 and 2 as “Huge” and “Very large” effects, respectively.

## 3. Results

### 3.1. Baseline Characteristics

The study population included diabetic subjects with a mean age of 54 years and a BMI of 26.9 kg/m^2^, and the Control group included individuals with a mean age of 41 years and a BMI of 26 kg/m^2^. The D group consisted of 57 males and 45 females, the DG group consisted of 49 males and 55 females, and the Control group consisted of 62 males and 42 females.

The baseline characteristics of subjects in each group are presented in Table 1. Concentrations of FPG, PPG, FPI, HbA1c, and 8-OHdG were significantly high and that of GSH was significantly low in D and DG compared to Control (*p* < 0.001, all parameters). Levels of PPI in D and DG were not found to be significantly different compared to the Control group (Table 1).

### 3.2. Oral GSH Supplementation Increases Erythrocyte GSH and Decreases Oxidative Damage to DNA but Does Not Alter Glycemia in Diabetic Patients over a Period of Six Months

GSH levels increased significantly over a period of six months, from the α to γ visit in both DG (*p* < 0.001) and D (*p* < 0.001) groups, while they remained unchanged in the Control. We further estimated the effect size of GSH supplementation within the diabetic groups: A “Large” effect (Cohen’s d = 1.01; *p* < 0.001) indicated that the increase in GSH is significantly high in DG compared to D (Figure 2). GSSG was similarly increased in DG compared to D (Cohen’s d = 0.61, *p* < 0.001).

We also observed a significant decrease in the concentrations of 8-OHdG from the α to γ visit with a “Large” effect in DG (Cohen’s d = −1.07; *p* < 0.001) but not in the D and Control groups (*p* > 0.05) (Figure 2).

We then analyzed the effect of oral GSH supplementation on the glycemic parameters in diabetic patients. We observed that HbA1c levels decreased significantly over six months in both D and DG; however, the extent to which it decreased in DG was comparable to the D group, as indicated by a small Cohen’s d = −0.16 (*p* > 0.05) (Figure 2). FPG, PPG, FPI, and PPI decreased over a period of six months in D and DG; however, changes in DG were comparable to those in D (*p* > 0.05, Cohen’s d < 0.2, all parameters).

Overall, our results indicate that GSH supplementation leads to a significant increase in the erythrocyte GSH and GSSG and a decrease in 8-OHdG in diabetic patients. However, the changes in the glycemic parameters of D and DG were to similar extents.

Next, we investigated whether the effect of GSH supplementation is accomplished rapidly and stabilized thereafter, or whether their levels change gradually over a period of six months (Appendix A).

### 3.3. Oral GSH Supplementation Enhances Erythrocyte GSH in Diabetic Subjects within Three Months

Figure 3a,b show serial changes in the concentrations of GSH and GSSG, respectively, from the α to β and γ visits in the three study groups. In Control, GSH and GSSG remained unchanged over a period of six months. GSH supplementation in DG led to a significant increase in GSH within the first three months (*p* < 0.001) and remained stable thereafter for up to six months (Figure 3a). In the D group, on the other hand, GSH increased marginally from 0 to 3 and 6 months. In the DG group (Figure 3b), GSSG also increased significantly within the first three months (*p* < 0.001), and did not change further. In D, GSSG remained unchanged during the study period. Thus, oral GSH supplementation in diabetic patients increased GSH significantly within three months and stabilized it thereafter. On the other hand, in D, anti-diabetic therapy alone led to a small increase in GSH.

### 3.4. Oral GSH Supplementation Significantly Reduces 8-OHdG in Diabetic Subjects

In the Control group, concentrations of 8-OHdG remained unchanged over a period of six months, while GSH supplementation in diabetic patients led to a significant decrease in 8-OHdG within the first three months, which continued to reduce significantly thereafter (*p* < 0.001) (Figure 4a). However, in the D group, its concentrations did not change significantly.

### 3.5. HbA1c Levels Are Stabilized by Oral GSH Supplementation in Diabetic Patients

We examined serial changes in the levels of glycemic parameters in D and DG groups in greater detail. FPG levels lowered significantly within three months in D (*p* < 0.01) and DG (*p* = 0.05); however, they recovered to the baseline levels by the end of six months (Figure 4b). PPG levels, on the other hand, did not change significantly in D and DG over a period of six months (*p* > 0.05, all) (Figure 4c). HbA1c rapidly decreased from 0 to 3 months in both D (*p* < 0.01) and DG (*p* < 0.001) (Figure 4d). Thereafter, HbA1c levels were maintained until 6 months in DG, while they appear to have returned to the baseline in the D group.

FPI levels changed significantly from 0 to 3 and 6 months in D (*p* < 0.05), while they remained unchanged in DG (*p* > 0.05) (Figure 4e). PPI levels remained unaltered in D and DG throughout the study period (Figure 4f). Taken together, oral GSH supplementation in diabetic patients appears to have a “stabilizing effect” on HbA1c, i.e., it decreases rapidly within three months and continues thereafter.

### 3.6. A Oral GSH Supplementation Significantly Reduces HbA1c in Elderly Diabetic Patients

Earlier reports suggest that the concentration of GSH decreases with age in healthy adults [17,26]. Therefore, we assessed the effect of GSH supplementation in elderly diabetic patients.

Diabetic patients in our study ranged from 31 to 78 years of age. The median age in these cohorts is about 55 years in the D and DG groups. We used this age as a threshold to isolate an elder sub-group. We then re-examined the effect of GSH supplementation in this sub-group diabetic population to assess whether they respond differently to oral GSH supplementation compared to the younger population.

Mean values for all the biochemical parameters and serial changes from 0 to 3 and 6 months in their concentrations in the D (*n* = 44) and DG (*n* = 54) groups are shown in Appendix A, respectively. Similar to results obtained for diabetic patients overall, the concentration of GSH and GSSG increased significantly over a period of 6 months in both the D and DG sub-groups (Appendix A). Changes in the mean GSH and GSSG over a period of 6 months in the DG group (Figure 5) were significantly higher compared to the D group (Cohen’s d = 1.14 and 0.67 for GSH and GSSG, respectively, *p* < 0.001).

GSH supplementation also resulted in a “Very large” effect (Cohen’s d = −1.45, *p* < 0.001) in the reduction of 8-OHdG in the elderly sub-group of diabetic patients (Figure 5), suggesting that oral GSH supplementation in the elderly diabetic population results in a significant reduction in the accumulation of oxidative DNA damage.

Next, we examined the effect size of blood glycemic parameters in response to oral GSH supplementation in the elderly sub-group of diabetic patients (Figure 5). In contrast to the results observed in the diabetic population overall, GSH supplementation in the DG sub-group led to a significant reduction in HbA1c over a period of 6 months compared to D (Cohen’s d = −0.41, *p* < 0.05).

Interestingly, FPI levels also increased significantly in the DG sub-group from the α to γ visit compared to D (Cohen’s d = 0.56, *p* < 0.05). GSH supplementation had a small effect on levels of FPG, PPG, and PPI in the DG sub-group (Cohen’s d < 0.2, *p* < 0.05, all parameters).

Thus, GSH supplementation in the DG sub-group of elderly diabetic patients over a period of 6 months led to a significant increase in the erythrocyte GSH, GSSG, and FPI and a decrease in HbA1c and 8-OHdG levels (Appendix A), suggesting that the elderly diabetic population responds better to GSH supplementation in conjunction with anti-diabetic therapy. The effect of GSH supplementation has been also analyzed in the younger sub-groups of D and DG and the results are shown in the supplementary documents (Appendix A).

HbA1c levels changed significantly from the baseline in the younger sub-group of DG (Appendix A); however, the 6-month changes in the younger sub-group of DG did not show any significant difference when compared to the 6-month changes in the younger sub-group of D as a result of GSH supplementation (Appendix A).

## 4. Discussion

GSH, a water-soluble tri-peptide, is an important endogenous antioxidant required for maintaining the redox homeostasis of the cell. It is synthesized by glutathione synthetase and utilized by glutathione peroxidase and glutaredoxin to detoxify free radicals. Several studies have reported low levels of GSH in different pathological conditions [6]. GSH insufficiency can be due to increased exposure to oxidants, drugs, excess nutrients, or decreased rate of synthesis of GSH. In our earlier studies, we found a significant decrease in GSH in T2D individuals, and among 12 different markers of OS measured, GSH impressively correlated with changes in HbA1c [27], suggesting that altering hyperglycemia rapidly results in changes in GSH.

Interventions aiming at controlling hyperglycemia are the primary line of treatment for diabetic patients. It is interesting to ask if improving redox status by GSH supplementation can help counteract the deleterious effects of hyperglycemia-induced OS. Results from earlier clinical trials of oral GSH supplementation have been contrasting and debatable. Our study provides clear evidence that long-term oral GSH supplementation not only improves body stores of GSH but significantly decreases the accumulation of oxidative DNA damage in Indian T2D patients. It also helps increase the efficiency of anti-diabetic treatment in maintaining normoglycemia in diabetic patients.

GSH is known to be either transported in its intact form from the intestinal epithelial cells into the blood lumen [28] or broken down by gamma-glutamyl transferase to its constituent amino acids [29]. It is unclear whether GSH was either directly absorbed or broken down into its constituent amino acids and re-synthesized by glutathione synthetase. Additionally, we find a significant increase in the concentration of erythrocytic GSSG. This is possibly due to the conversion of erythrocytic GSH into GSSG, in line with previous reports; for instance, Nolan et al. [28] show that ^13^C-GSH administered to mice is rapidly converted to GSSG and accumulated in red blood cells and liver. Thus, oral GSH supplementation not only increases body stores of GSH but a fraction is stored as GSSG. These results strongly suggest that GSH supplementation results in a systemic improvement of the redox state in diabetic individuals. The augmentation of antioxidant reserves, by elevating GSH levels, also resulted in a significant reduction in the accumulation of oxidative DNA damage implicated in the pathophysiology of diabetic complications.

HbA1c levels typically fluctuate despite regular anti-diabetic treatment in diabetic patients. We found that GSH supplementation helped maintain lowered HbA1c within three months. This effect was more pronounced in elderly patients over 55 years of age. Other characteristics of the glycemic state, such as FPG, PPG, FPI, and PPI, did not change in the diabetic patients overall; however, interestingly, we observed an increase in FPI levels in elderly diabetic patients. The exact mechanism by which GSH helps in maintaining normoglycemia in diabetic patients requires further investigation.

Preserving β-cell function is essential to glucose control in diabetic patients. It is crucial to maintain a healthy redox state of pancreatic β-cells, as their ability to secrete insulin in response to glucose is dependent on intracellular thiols [30]. It is well established that β-cells are more vulnerable to ROS due to their low antioxidant capacity and poor ability to repair oxidatively damaged DNA [22,31]. Thus, one potential strategy for improving β-cell function is to provide antioxidant support to pancreatic β-cells. Enhancing extracellular GSH levels improved β-cell response to glucose in rats [32]. Infusion with GSH [12] also enhanced β-cell function and consequently improved glucose disposal in patients with impaired glucose tolerance. Our results also indicate that oral GSH supplementation supports anti-diabetic treatment in reducing hyperglycemia. It is difficult at this stage to establish a causal sequence of events that underlie these observations. For instance, while it is generally believed that the etiology of diabetes in Southeast Asian diabetic patients points to especially poor insulin resistance [33], recent reports have indicated that a large sub-group of patients belong to an insulin-deficient phenotype [34]. We note, in particular, that HOMA indices were only one component of a more comprehensive clustering pattern that included age at diagnosis, HbA1c, HOMA2-ß, HOMA2-IR, and BMI. We further point out that there are there few patients in this study (*n* = 7) who were on insulin, hence it is unclear if the observations described here extend to those in whom insulin insufficiency is severe.

It is known that concentration of GSH declines with aging [17,26] and this could be further aggravated in elderly diabetic patients. We indeed observed that elderly diabetic patients benefited more from GSH supplementation both in terms of reducing oxidative DNA damage and improving glycemic status. Interestingly, we also observed a significant increase in FPI in these elderly patients. Recently Zhang et al. [35] reported restoration of β-cell function by administration of oral GSH in diabetic rats. Islets isolated from T2D cadaveric organ donors showed impaired insulin secretion in response to glucose and increased levels of oxidative damage markers. Treating these islets with GSH led to an improvement in their functionality and also alleviated oxidative damage markers [36], suggesting that reducing OS in islets could be a potential target for treating diabetes. We speculate that a systemic increase in GSH in diabetic patients resulted in a significant reduction in oxidative DNA damage, improved the pancreatic β-cell function, and concomitantly reduced HbA1c, prominently so in elderly diabetic individuals. However, these results need to be further validated in large clinical settings.

T2D is a multifactorial, complex disease and can be controlled by diet modifications, control of physical activity, weight reduction, etc. These factors need to be considered for the personalization of therapy. It would also be interesting to see how long the effect of GSH intervention persists; since the antioxidant status of an individual varies widely, it is plausible that this can significantly influence the effect of exogenous supplementation. This might even explain why the changes in HbA1c observed in DG have shown limited effect sizes. It is also conceivable that longer intervention with GSH may show further improvements in glycemic parameters, such as fasting glycemia. In our study, due to sample size limitations, we do not have enough statistical power to perform such analyses. However, our work lays the foundation for further studies with various population cohorts to understand these effects better.

Our results have provided support for significant, if modest, effects of GSH supplementation on HbA1c. This is very important, especially in light of the ADA position [37], which recognizes that the personalization of anti-diabetic therapy—rather than a one-size-fits-all treatment—is necessary to achieve successful glycemic targets. However, few algorithms exist that describe how to achieve this ambitious goal. For this reason, we reiterate that GSH supplementation is an important addition to this toolbox. We have shown significant positive benefits of GSH, and importantly, it is tolerated very well by patients; this makes it a very useful therapeutic agent to add to the clinician’s arsenal.

## 5. Conclusions

Our results strongly suggest that oral GSH supplementation replenishes the body’s stores of GSH and significantly reduces oxidative DNA damage in Indian T2D patients. It also reduces HbA1c within three months and maintains it thereafter in the diabetic population overall. An elderly sub-group seems to benefit greatly, as evidenced by a significant decrease in HbA1c and an increase in insulin secretion by β-cells over a period of six months. A clinical implication of our study is that the oral administration of GSH can be used as an adjunct therapy to anti-diabetic treatment in achieving better glycemic targets, especially in the elderly population.

## Figures and Tables

**Figure 1 antioxidants-11-01026-f001:**
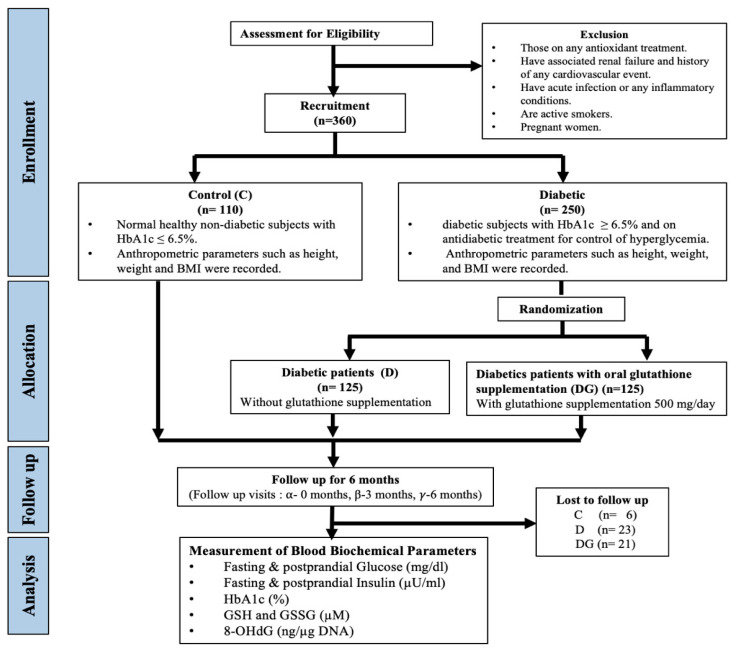
Flowchart for study design.

**Figure 2 antioxidants-11-01026-f002:**
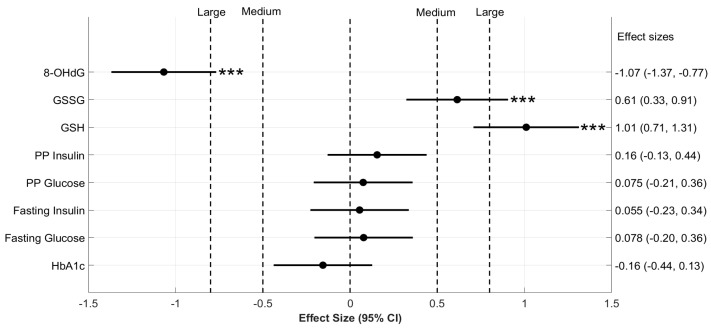
**The effect size of changes in blood biochemical parameters.** Six-month changes in the biochemical parameters of D and DG groups were compared here on a forest plot with effect size and corresponding 95% confidence intervals (CI). Effect size (Cohen’s d) calculated between 6-month changes in the concentration of biochemical variables are denoted on the *x*-axis. The group-wise means of 6-month changes in the concentration of these variables were compared using two-sample permutation tests. The significance of these comparisons is denoted by the p values mentioned to the right of horizontal lines for CI. Significance level is *** *p* < 0.001 for respective comparisons. Effect size takes either a positive or negative sign based on the direction of change: a positive effect size increases towards the right and a negative effect towards the left. Vertical dotted lines represent different classifications of effect size. In particular, “Medium” effects are labeled at 0.5 and −0.5, and “Large” effects at 0.8 and −0.8. Abbreviations used here are, HbA1c—glycated hemoglobin, GSH—reduced glutathione, PP glucose—postprandial glucose, PP insulin—postprandial insulin, and 8-OHdG—8-hydroxy-2-deoxy guanosine.

**Figure 3 antioxidants-11-01026-f003:**
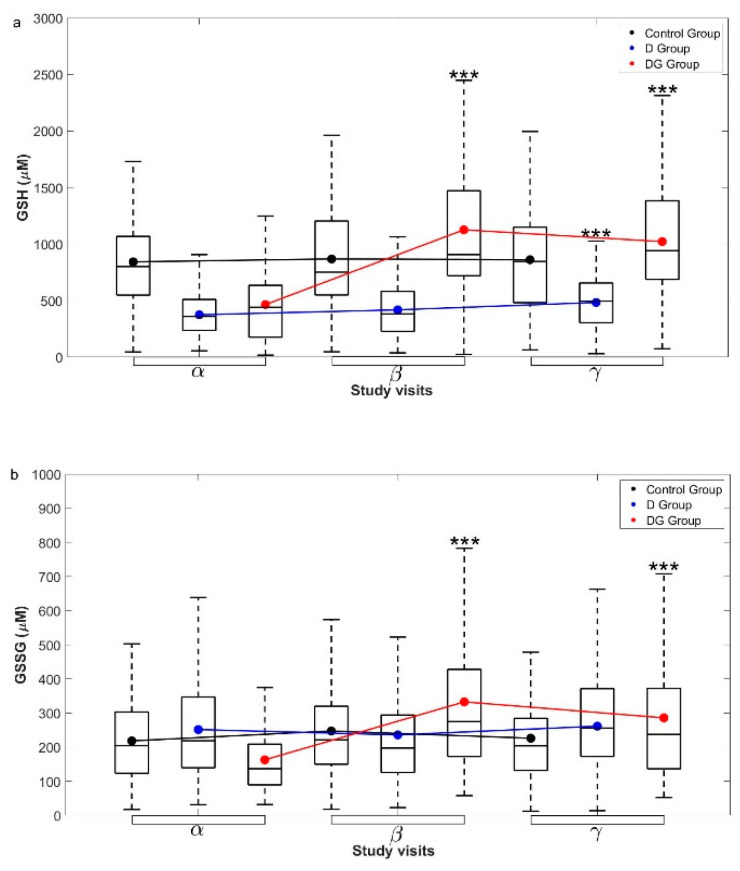
**Longitudinal changes in the concentration of (a) GSH and (b) GSSG in different groups.** The measured data for (**a**) GSH and (**b**) GSSG concentrations from Control, D, and DG groups at α, β, and γ visits are shown here with box and whiskers plots. Mean (black circles for Control, blue for D, and red for DG groups, respectively) and inter-quartile ranges (IQR) of the data are overlaid over the corresponding box plots. The group-wise means at different visits are connected using solid lines with the same color. Significance levels displayed above β, and γ visits denote the comparisons with α visit using permutation tests. Significance level is *** *p* < 0.001 for respective comparisons. Abbreviations used here are, GSH—reduced glutathione, and GSSG—oxidized glutathione.

**Figure 4 antioxidants-11-01026-f004:**
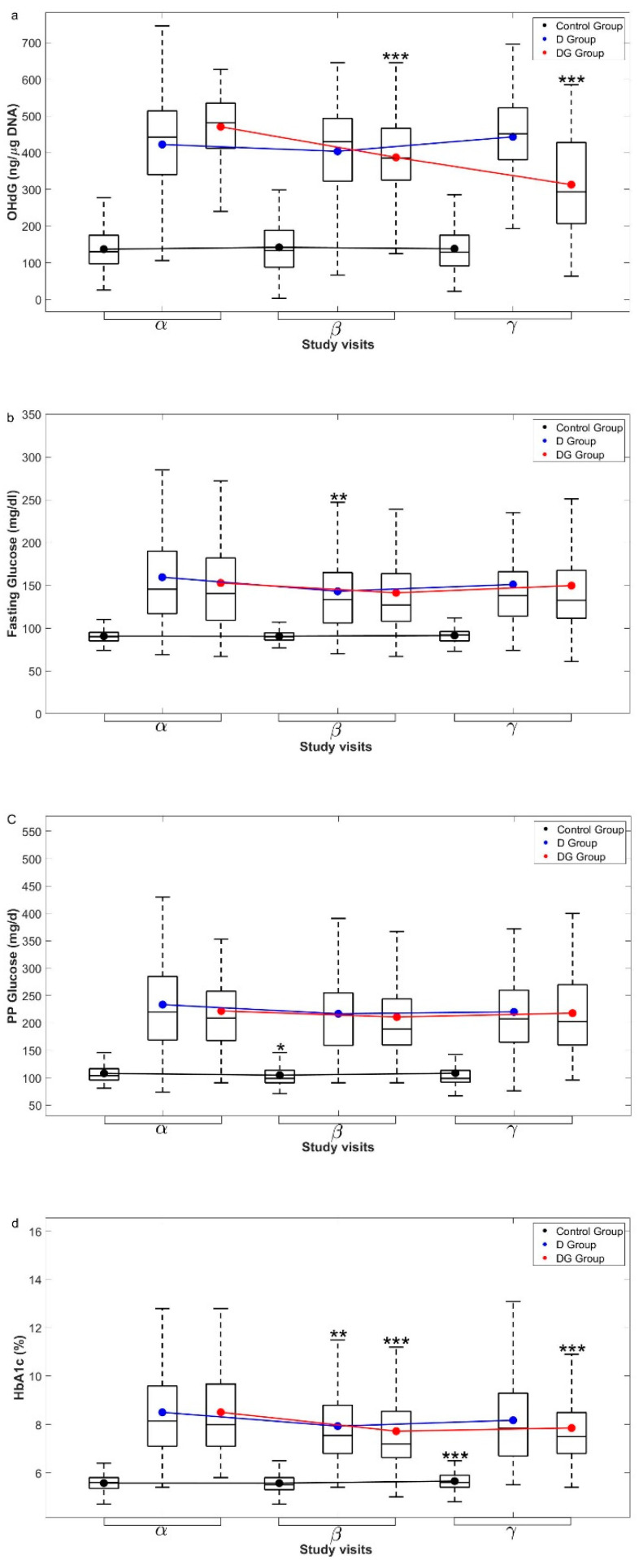
**Longitudinal changes in glycemic parameters**. The measured data for (**a**) 8-OHdG, (**b**) fasting glucose, (**c**) PP glucose, (**d**) HbA1c, (**e**) fasting insulin, and (**f**) PP insulin concentrations from Control, D, and DG groups at α, β, and γ visits are shown here with box and whiskers plots. Mean (black circles for Control, blue for D, and red for DG groups) and IQR of the data are overlaid over the corresponding box plot. The group-wise means at different visits of a group are connected using solid lines with the same color. Significance levels (*) displayed above β, and γ visits denote the comparisons with α visit using permutation tests. Significance levels are * *p* < 0.05, ** *p* < 0.01, and *** *p* < 0.001 for respective comparisons. Abbreviations used here are, 8-OHdG—8-hydroxy-2-deoxy guanosine, PP glucose—postprandial glucose, HbA1c—glycated hemoglobin, PP insulin—postprandial Insulin.

**Figure 5 antioxidants-11-01026-f005:**
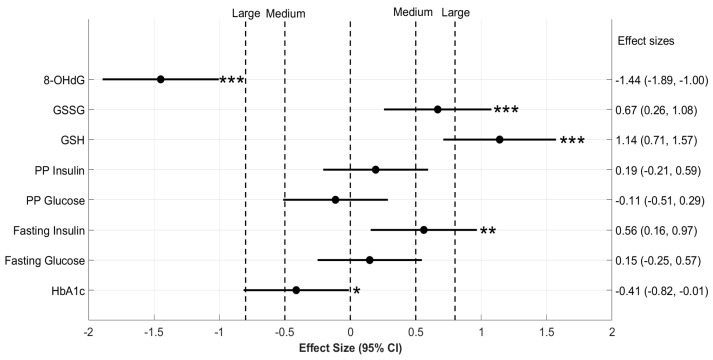
**The effect size of changes in blood biochemical parameters of elderly diabetic patients.** Six-month changes in the biochemical parameters of those in D and DG sub-groups were compared here on a forest plot with effect size and 95% confidence intervals. Effect size (Cohen’s d) calculated between 6-month changes in the concentration of biochemical variables are denoted on the *x*-axis. The group-wise means of 6-month changes in the concentration of these variables were compared using two-sample permutation tests. The significance of these comparisons is denoted by the p values mentioned to the right of horizontal lines for CI. Significance levels are * *p* < 0.05, ** *p* < 0.01, and *** *p* < 0.001 for respective comparisons.

**Table 1 antioxidants-11-01026-t001:** **Baseline characteristics of Control, D, and DG groups.** Data from each group at the α visit are presented here as median and IQR, inter-quartile ranges (25th percentile–75th percentile). * indicates the significance of the comparison between baseline measurements of Control versus D or Control versus DG groups. Significance levels are * *p* < 0.05, ** *p* < 0.01, and *** *p* < 0.001. Similarly, significance levels for comparisons between D versus DG groups are denoted with ^##^, or ^###^ for *p* < 0.05, *p* < 0.01, and *p* < 0.001, respectively. We did not observe any significant differences in the levels of FPG, PPG, FPI, PPI, HbA1c, and GSH within the D and DG groups, thus confirming covariate balance in the two groups at baseline (Appendix A). Abbreviations used here are BMI—body mass index, HbA1c—glycated hemoglobin, GSH—reduced glutathione, PP glucose—postprandial glucose, PP insulin—postprandial insulin, and 8-OHdG—8-hydroxy-2-deoxy guanosine.

Biochemical Variables	Control ----------------- Median (25th–75th Percentile)	D ----------- Median (25th–75th Percentile)	DG ---------- Median (25th–75th Percentile)
Age (years)	39.5 (33.5–49)	55.5 (47–61) ***	56 (48–61) ***
BMI (kg/m^2^)	26.1 (23.5–28.2)	26.3 (22.7–29.2)	26.8 (23.8–29.8)
HbA1c (%)	5.6 (5.4–5.8)	8.1 (7.1–9.6) ***	8 (7.1–9.7) ***
Fasting Glucose (mg/dL)	90 (85–95)	147 (120–190) ***	140.5 (109–182) ***
Fasting Insulin (µU/mL)	9.4 (6.8–12.3)	11.9 (7.4–17.1) **	10.4 (7.5–16.1) *
PP Glucose (mg/dL)	104 (96–117)	220 (169–285) ***	209 (168–258) ***
PP Insulin (µU/mL)	36 (18.1–71.7)	36.2 (24–54.8)	32.4 (18.1–60.4)
GSH (µM)	801 (548–1068)	379 (243–533) ***	440 (176–635) ***
GSSG (µM)	205 (124–303)	215 (139–326)	137 (89–209) ***^,###^
8-OHdG (ng/µg DNA)	129.97 (97.2–175.2)	442.33 (340.26–514) ***	481.71 (412.23–535.11) **^,##^

## Data Availability

All data generated or analyzed during this study are included in this published article, including supplementary files, and can be downloaded from https://figshare.com/s/0803267e1d38c054cee6 (accessed on 24 August 2019). Please note that the folder also contains scripts that can be used to conveniently reproduce the analysis in the manuscript.

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
