# Peer review of "Randomized Clinical Trial of How Long-Term Glutathione Supplementation Offers Protection from Oxidative Damage and Improves HbA1c in Elderly Type 2 Diabetic Patients"

_antioxidants, 2022, doi:10.3390/antiox11051026_

Round 1

Reviewer 1 Report

All my previous comments have been addressed and I do not have further comments.

Reviewer 2 Report

Authors have satisfactory answered the comments indicated by the Reviewer. The manuscript is accepted in te present form.

Reviewer 3 Report

In this new submission, authors did not addressed the issues previously raised by this reviewers.

Reviewer 4 Report

Although the Authors responded to the comment they did so incorrectly. By centrifuging the blood at the given speed, the authors separated all blood cells and platelets from the serum/plasma. It is not possible to separate the remaining cell fractions from the erythrocytes only by centrifuging, the authors could only do so by centrifuging in a gradient or using a cell sorter. It is not a methodological error to use the thus obtained cell fraction for GSH and GSSG determinations, but such a formulation is misleading the reader. Please change to "Blood samples were centrifuged at 4000 rpm for 10 mins to separate morphological elements from whole blood. Plasma was stored at -80 ° C (Thermo Fisher Scientific, USA) for further analysis."

This manuscript is a resubmission of an earlier submission. The following is a list of the peer review reports and author responses from that submission.

Round 1

Reviewer 1 Report

This study demonstrated through a well-designed clinical trial how the administration of glutathione to patients with type 2 diabetes already on treatment, in addition to improving body glutathione stores and reducing oxidative stress, improved Hb1Ac control, especially in older subjects.

In my view the following point should be addressed to complete this study:

What happens when were only younger diabetic participants examined? Maybe, in younger patients, glutathione administration did not have any positive effect in the maintenance of decreased Hb1Ac levels … Thus, the same analysis done in elderly patients (supplemental figure 1 and supplemental figure 2) should be performed in younger diabetic patients, and included these data as a supplemental figure 3 and 4, and in the text of Results section.

Minor comment:

In table 1, the mean of ## or ### should be detailed in table footnotes.

Reviewer 2 Report

Manuscript 1665637, sent to ANTIOXIDANTS

Randomised clinical trial of long-term glutathione supplementation offers protection from oxidative damage, improves HbA1c in elderly type 2 diabetic patients

By: Saurabh Kalamkar, Jhankar Acharya, Arjun Kolappurath Madathil, Vijay Gajjar, Uma Divate, Sucheta Karandikar-Iyer5, Pranay Goe, Saroj Ghaskadbi

The study aimed to study the long-term effect of glutathione supplementation (500 mg) on different parameters related to oxidative stress and glycemic control in type-2 diabetic patients. Previous studies in this field have been carried out with a low number of participants with a variety of glutathione doses and treatment duration. For this reason, clear conclusions were difficult to reach. The present report addresses the problem in a large population of type-2 diabetic patients, showing more conclusive results. The statistics used seem appropriate and results are correctly presented. Nevertheless, interpretation of results seems very optimistic and the authors need to give more realistic conclusions from this interesting study. Here are some points that need additional information or changes in the interpretation.

INTRODUCTION: The relation of the pathology with oxidative stress is well documented. A key point is the administration of GSH to patients in the different studies. This is a key point regarding body GSH levels during supplementation. Authors can present some data regarding which form of GSH administration seems to work better:  sublingual, orobuccal, liposomal, precursor amino acids. Authors should present a range of doses used in the different published studies as well as a range of duration of the different interventions. This information will help the reader to understand the protocol design of the present study in terms of form of GSH administered, dose and length of supplementation.

MATERIALS AND METHODS: Following with the previous criticism, authors must indicate: the form of GSH administered. Information regarding the treatment with anti-diabetic drugs must be provided. Some of the available drugs could have a certain anti-oxidant capacity. It would be interesting to know if patients taking certain drugs display significant improvements in the glycemic control and in Hb1Ac, as well as insulin levels. In addition, the diet and level of physical activity followed by participants was similar to the control group? These variables have a clear influence in oxidative stress and it is possible that GSH supplementation works better in patients following an antioxidant rich diet and performing programmed physical activity than patients that do not accomplish these items. The answer to all these points will help to validate all comparisons between the 3 experimental groups (C, D and DG).

RESULTS: Results are well presented in the manuscript as well as in the supplementary file. The interpretation why supplementation works better in elders seems to be reasonable. Supplementation depends of the sex? Works better in men or in women? Please give some information.

DISCUSSION: The main concern refers to the optimistic interpretation made by the authors. GSH supplementation improves antioxidant content in DG group. However the changes in Hb1Ac and fasting glycaemia are still very modest. It is true that they are significant but differences comparing to the beginning of the study are small and out to the healthy range. Authors should discuss this point indicating improvements to complete GSH supplementation for future interventions: diet modifications, control of physical activity, weight reduction, length of intervention, etc. In addition, the presented results suggest that patients undergo an insulin resistance rather than a decrease in beta-cell mass due to glucolipotoxicity. They have to mention this point indicating that in this diabetes stage antioxidants interventions seem to be more effective than facing a more stablished pathology needing insulin injections. It is important to inform why these therapeutic actions are more effective at this stage.

Reviewer 3 Report

This manuscript is potentially interesting, but it suffers from severe methodological limitations.

1- At baseline the metabolic conditions of the groups were significantly different. Therefore the authors' conclusions are conditioned by an uncorrectable bias.

2- Individuals over the age of 55 cannot be defined as elderly, neither from a clinical nor a biological point of view.

3- A final section of the discussion is missing in which the authors comment on the limitations of the study

Reviewer 4 Report

The manuscript entitled „ Randomised clinical trial of long-term glutathione 2 supplementation offers protection from oxidative damage, 3 improves HbA1c in elderly type 2 diabetic patients “describes the clinical implication of oral administration of GSH potentially complements anti-diabetic therapy in achieving better glycemic targets, especially in the elderly population. The manuscript is interesting and relatively well written. However, I have a few minor suggestions which should be improved before publication.

Minor

  • Line: 148 “separate plasma and erythrocyte fraction.” The authors separated all types of cells present in the blood from the serum, not only erythrocytes, the most abundant fraction.
  • Abbreviations should also be developed for the first time in the manuscript’s text, not only in the abstract, e.g., T2DM.
  • Suggestion: an excellent complement to the research would be to add the calculated indicators of insulin resistance (HOMA IR) or insulin sensitivity (Quicki) and or McAuley, but for the last one, it is necessary to provide changes in TG in blood serum
  • For the sake of the readers, I propose to add an abbreviation index
  • Although I am not a native speaker and I cannot say the correctness of English, it seems that the paper’s language needs to be improved by a professional language company or a native English-speaking person.
  • The methodology should be described in more detail not only through references to methods in other publications.